# Integrating Mental Health Services into Primary Care Settings: A Multiple Case Study of Congolese Experiences Testing the Feasibility of the WHO’s Mental Health Gap Action Programme

**DOI:** 10.3390/ijerph22030457

**Published:** 2025-03-20

**Authors:** Erick Mukala Mayoyo, Bart Criel, Aline Labat, Yves Coppieters, Faustin Chenge

**Affiliations:** 1School of Public Health, Université Libre de Bruxelles, 1070 Brussels, Belgium; aline.labat@ulb.be (A.L.); yves.coppieters@ulb.be (Y.C.); 2School of Public Health, University of Lubumbashi, Lubumbashi 1825, Democratic Republic of the Congo; fchenge@hotmail.fr; 3Section de Santé Communautaire, Institut Supérieur des Techniques Médicales de Kananga, Kananga 321, Democratic Republic of the Congo; 4Centre de Connaissances en Santé en RD Congo, Kinshasa 3088, Democratic Republic of the Congo; 5Department of Public Health, Institute of Tropical Medicine in Antwerp, 2000 Antwerp, Belgium; bcriel@ext.itg.be

**Keywords:** integration, mental health, primary healthcare, mental health Gap Action Programme, multiple case study, Democratic Republic of Congo

## Abstract

Some experiences of integrating mental health into primary care settings, testing the feasibility of the World Health Organization’s mental health Gap Action Programme, have been launched in the Democratic Republic of the Congo to address treatment gaps. However, they have not yet been documented to look at scaling up. This study described the health outcomes and lessons learned from two of these experiences. A multiple case study was conducted on two integration programmes in the urban Tshamilemba district in the Haut-Katanga province, ongoing since 2021, and in the rural Mangembo district in the Kongo Central province, ongoing since 2022. Data were collected between July and August 2024 from focus group discussions, interviews, document reviews, including routine health information systems. We carried out descriptive statistical analyses to measure indicators of accessibility and the use of services, and content analysis to explore the lessons learned. A total of 1708 individuals with mental disorders were treated in primary care settings between 2021/22 and 2024 under both programmes. From 2021 to 2024, the curative consultations rate for mental disorders, which was unknown at the start of both programmes, reached 14.4 new cases/1000 inhabitants/year in the Tshamilemba district and 14.2 new cases/1000 inhabitants/year in the Mangembo district. Several lessons were learned, related to each phase of mental health Gap Action Programme. The findings confirm the feasibility and effectiveness of the mental health Gap Action Programme in the Congolese context and highlight the need for concerted action to address the identified challenges.

## 1. Introduction

The burden of mental health disorders remains high in low-income countries (LICs), including the Democratic Republic of the Congo (DRC). In 2021, 14.2 million people were reported to have mental health disorders, with 1.69 million disability-adjusted life years (DALYs), and an estimated prevalence of 13.5%, making these disorders the seventh leading cause of morbidity in the DRC [1,2]. During the same period, at least 13,020 suicide deaths and 260,400 suicide attempts were reported [2]. Considering the burden of neurological disorders (DALYs per 100,000 inhabitants: 946.7) and those linked to substance use (DALYs per 100,000 inhabitants: 171.4) [1], this burden of disease increases considerably. As healthcare financing in the DRC is mainly provided by households through out-of-pocket payments [3], the cost of treatment and care for mental health disorders would have a considerable negative impact on families and the national economy. It has been reported that in the DRC, for an episode of mental health disorder lasting 6 months, patients would pay up to USD 3600 in a psychiatric hospital [4].

Despite this huge epidemiological and economic burden of mental health disorders, there are still large treatment and care gaps in the DRC, posing major challenges to making substantial progress towards the goal of “health for all”. The coverage of mental health services in the “formal” healthcare system is estimated at 5% nationwide [5]. Healthcare facilities delivering mental health services remain essentially specialised and centralised and are sorely lacking at the peripheral level of the health system, i.e., health districts. To date, the country has only six recognised psychiatric hospitals with a total capacity of 500 beds, some 30 faith-based mental health centres, and private mental health clinics [6,7,8], for a population estimated at 122 million in 2024 [9], with most of them being based in the capital Kinshasa. In addition, financial, material, and human resources for mental health services are inadequate. For instance, the country has 0.1 psychiatrists, 0.25 mental health nurses, and 0.02 clinical psychologists per 100,000 inhabitants [8], indicating the major challenges to be addressed if the goal of ‘health for all’ is to be achieved.

Given that the provision of biomedical mental health services remains almost non-existent in primary care settings [10,11], the majority of people in the DRC do not have access to evidence-based mental health services. Around 80% of people with unmet mental healthcare needs rely mainly on traditional approaches to care, including traditional medicine, prayers, incantations, and other alternative and informal care approaches, in both urban and rural areas [10]. Thus, there is a need for an urgent adoption of strategies to address gaps in mental health services. Therefore, the DRC, which has an ambitious national mental health plan [12], plans to address these mental healthcare gaps for all Congolese, by improving access through primary healthcare (PHC) settings.

In the DRC, PHC settings still have some way to go before they can offer quality services, including mental health services. When caring for people with mental, neurological, and substance use (MNS) disorders, it is currently very difficult in the country for PHC settings, which are consulted by around 80% of the population [13], to correctly identify common MNS disorders and comorbidities, provide appropriate holistic curative care, ensure optimal rehabilitation for people with psychosocial disabilities, and provide appropriate mental health prevention and promotion, including the reduction in mental health-related stigma, etc. A pragmatic solution to address treatment and care gaps is to improve access to mental health services in primary care settings by shifting to or sharing mental health tasks with non-specialist care providers [14,15,16]. To achieve this, the World Health Organisation (WHO) recommended integrating mental health services into PHC systems using evidence-based and cost-effective approaches [17]. In our previous research, we defined integration as the process of introducing a mental health services package into the health activities of existing primary care facilities (i.e., health centres and the district hospital) and at the community level in a health district [18].

To facilitate the implementation of integration in resource-limited countries, the WHO [19] developed the mental health Gap Action Programme (acronym ‘mhGAP’) [19]. The mhGAP emphasises the relevance of a multidisciplinary approach and intersectoral collaboration (including the health, social, educational, nutritional, and communication sectors, etc.) in addressing mental health problems, including community mental health [20]. The mhGAP is a guideline developed in 2008. It is supported by its intervention guide, i.e., the mhGAP-IG. The mhGAP operations manual describes the three-phase mental health integration process and its main activities, which are listed below (Box 1) [20,21,22].

Since 2011, when a pilot integration experience was launched in the east of the country [5], the DRC’s National Mental Health Programme (PNSM) has adopted the WHO’s mhGAP. Since then, a number of attempts to partially integrate mental health into the PHC system are being tested. To date, two ongoing experiences have attracted our attention, as both use the WHO’s mhGAP. These are the integration programme (or experience) in the Tshamilemba health district, in Haut-Katanga province, covering the period from May 2021 to December 2025; and the integration programme in the Mangembo health district, in the province of Kongo Central, covering the period from December 2022 to December 2026. The Tshamilemba integration programme is funded by the Institute of Tropical Medicine (ITM) in Antwerp as part of its fourth framework agreement supported by the Belgian Directorate-General for Development Cooperation and Humanitarian Aid (DGD), while the Mangembo programme is financed by Memisa (a Belgian non-profit organisation) with funding from the DGD. In Tshamilemba, funding is limited to interventions relating to the integration of mental health, while in Mangembo, the district receives “systemic” financial support, spread over several years, for both the integration of mental health and its functioning. The aim of these two programmes is to demonstrate that the use of the WHO’s mhGAP for mental health integration in primary care settings in DRC’s health districts is feasible and effective. As such, these integration programmes would improve mental health outcomes, including access to and use of quality curative care among people with common MNS disorders attending PHC settings, and help reduce mental health-related stigma.

Box 1Phases and main integration activities recommended in the mhGAP operations manual [22].The mhGAP operations manual describes the mental health integration process in three following phases:Phase 1: Plan, which includes activities such as setting up an mhGAP operational team, conducting a situation analysis, developing an mhGAP operational plan and budget, and advocating for mental health;Phase 2: Prepare, which includes activities such as adapting the components of the mhGAP package, training staff in mhGAP, preparing for clinical and administrative supervision, coordinating care pathways, improving access to psychotropic drugs, and improving access to psychological interventions;Phase 3: Provide, which includes activities such as providing services at the facility level, providing treatment and care in the community, raising awareness of MNS conditions and available services, and supporting the implementation of prevention and promotion programmes.

It should be noted that although the country has a significant shortage of specialised mental health facilities, it does have at least 13,800 general healthcare facilities spread across 516 health districts [6], which could provide primary mental health services if this offer were integrated into them. This agrees with national sub-sectoral mental health policy [12]. This sub-sectorial mental health policy in the DRC was formulated in 1999 and revised in 2021 [12]. It aims to promote the mental health of the entire Congolese population by providing comprehensive, integrated, and continuous quality healthcare and services with community participation, in the overall context of the fight against poverty. Its vision is to ensure that all Congolese have equitable and sustainable access to quality mental health services and care defined in the (minimum and complementary) packages of activities, adjusted to the needs of users, taking account of contextual changes. The sub-sectoral metal health policy comprises seven strategic orientations: (i) the integration of mental health services and care packages into the health district and the continuity of care; (ii) the strengthening of leadership and governance; (iii) the development of human resources involved in mental health; (iv) the improvement of infrastructures and the availability of mental health equipment and materials; (v) the improvement of the availability of psychotropic medicines and supplies; (vi) the improvement of mental health funding; and (vii) the strengthening of intra- and inter-sectoral partnerships [12].

To the best of our knowledge, the two Congolese experiences testing the feasibility of the WHO’s mhGAP to integrating mental health into primary care settings, i.e., in the health districts of Tshamilemba and Mangembo, have not yet been rigorously documented. This raises doubts about the feasibility and effectiveness of the mhGAP in the Congolese context, making it difficult to argue in favour of scaling up the integration of mental health. In order to inform health system stakeholders, including decision-makers and their partners, on how to improve access to mental health services at health district level, the aim of this study was twofold: to describe the health outcomes of the two mental health integration programmes that tested the feasibility of the WHO’s mhGAP on the one hand, and on the other, to draw lessons learned from these experiences, to chart a pathway forward. This study would help fill evidence gaps on this topic and encourage (or not) the health system stakeholders to pursue efforts to integrate mental health.

## 2. Materials and Methods

### 2.1. Setting

This study was conducted in the urban health district of Tshamilemba in Lubumbashi and in the rural health district of Mangembo, both in the DRC. Tshamilemba is one of 27 districts in the province of Haut-Katanga, located in the southeast of the DRC, while Mangembo is one of the 31 districts in the province of Kongo Central, located in the western part of the country. Table 1 describes the main characteristics of these health districts at the start of the programmes U and R.

It should be noted that in the urban health district of Tshamilemba, only the single health centre of Tshamilemba, directly serving 12,000 inhabitants, was concerned with the integration of mental health, and was therefore the subject of the study [23]. Contrary to national health norms, this health centre has 15 doctors, 17 nurses, 2 midwives, and other health staff. In contrast, in the rural health district of Mangembo, all 19 healthcare facilities, i.e., 18 health centres and 1 district hospital in Mangembo, were involved [24].

### 2.2. Design

We conducted a multiple case study [25] on two similar experiences of integrating mental health services into PHC systems in Tshamilemba and in Mangembo, which are referred to as Programmes U (urban) and R (rural), respectively. Both programmes (i.e., U and R) involved interventions at three levels: the second level of care (district hospital) and the first level of care (health centre) as well as community. This collective, multiple “embedded” case study design, is therefore recommended when a study comprises more than one case and contains more than one sub-unit of analysis, thus making it possible to integrate qualitative and quantitative methods, with the triangulation of sources, data, analyses, etc., [26,27]. In addition, multiple case studies provide much more convincing results and evidence than single case studies, making this design more robust in the evaluation process when randomised studies with manipulation of variables are not possible [25].

### 2.3. Cases Selection

The selected cases are the two programmes for integrating mental health services into the PHC system (as defined above), ongoing in the urban health district of Tshamilemba and the rural health district of Mangembo. Table 2 summarises their main characteristics. The analysis units for these cases were the community, health centres in both Tshamilemba and Mangembo districts, and the Mangembo district hospital. The integration process, including its main activities, is defined by the WHO’s mhGAP operations manual described above. In the following paragraphs, we describe programmes U and R, and more in particular, the planned and implemented interventions.

### 2.4. Data Sources, Population, and Participant Selection

Data were collected from two sources: documents and key informants (KIs) who were (and/or still are) involved in the two integration programmes. The documents were drawn from three sources: (i) the health centre of Tshamilemba, a learning and research site of the School of Public Health of the University of Lubumbashi; (ii) the rural health district of Mangembo; and (iii) the DRC’s PNSM.

Three types of documents were selected purposively after applying the following criteria: documents dealing with the integration of mental health, written in French, dated from 2020 to 2024, and free of accessibility. After applying the above criteria, 12 documents: 3 project documents (narrative and statistical activity reports, etc.), 1 technical document, 1 policy document, as well as 4 of the routine health information system data files (.doc, .xlsx) were selected. These documents were complemented by 4 presentation files for the various workshops.

The KIs were healthcare providers, healthcare facility managers and health district managers team, (mental) health system decision-makers, specialist consultants in programmes, stakeholders in non-profit organisations active in mental health, academics and scientific experts in public (mental) health. They came from three countries: DRC, Guinea, and Belgium. A total of 37 KIs were selected on the basis of the following criteria: be over 18 years old; have been directly or indirectly involved in the design, planning and/or implementing of at least one of the two programmes studied; freely consent to take part in discussions or interviews and have a clear conscience.

### 2.5. Data, Collection Methods and Conducting a Multiple-Case Study

Qualitative and quantitative data were collected from July to August 2024. The qualitative data focused on the following: (i) stakeholders’ perceptions and opinions on the highlights of the integration programmes at the health centre of Tshamilemba and in the Mangembo district, in terms of the interventions planned and implemented, and on the changes in behaviour observed in reducing mental health-related stigma; (ii) the strengths and weaknesses, as well as the opportunities and threats of these programmes; (iii) possible solutions, recommendations and/or strategies for achieving greater sustainability; and finally, (iv) the lessons learned from the integration experiences testing the mhGAP guideline. The quantitative data concerned the intermediate quantitative outcomes of programmes obtained, in particular: the number of health professionals trained, the number of primary care facilities that have integrated mental health services provision, the number of people who have used health services, the number of patients who have experienced partial or total remission of symptoms, those who have dropped out of treatment, the number of patients referred and counter-referred in the healthcare system, etc.

To collect these data, we used different data collection methods: document review, including routine health information system review; focus group discussions (FGDs); and individual interviews. A combination of data collection methods is recommended for case study design [26,27]. Data collection was carried out in two phases.

First, we conducted a document review and routine health information system review from the health centre of Tshamilemba and the Mangembo district to collect the narrative and quantitative data contained in the documents. Second, we conducted three FGDs involving a total of 33 KIs; then 4 individual interviews with public (mental) health experts. The FGDs, organised in the form of workshops (face-to-face and online), and the individual interviews (exclusively online) were conducted in French, using an interview guide, and they lasted an average of 120 min. The FGDs and interviews were not recorded. However, a team of three research assistants (three male doctors with experience of qualitative health research) who had been previously trained took explanations and field notes of all the discussions.

During data collection, we used as a reference framework the theory of change (ToC) of the action research of Programme U. This ToC was inspired by the Programme for Improving Mental Health carE (PRIME). PRIME’s ToC has been used to improve access to mental healthcare in low- and middle-income countries (LMICs) [28]. The study procedure followed the operational model applied to the multiple case study [25].

At the end of the first analysis of the study documents, we discussed the preliminary results with two resource persons involved in the two programmes, who provided their assessment for a credibility check. We considered their comments during the second analysis of the documents. We triangulated the sources of information, the data in the documents, and the aggregated data provided by reporting, in order to minimise information bias.

### 2.6. Data Management and Ethics

The field notes and all explanations given during the FGD and/or interview were systematically reviewed before moving on to the next stage. This enabled us to identify how to (re)adapt questions in the next stages of data collection. During the transcription process, the data were anonymised and/or pseudonymised and managed as such after all participant identifiers had been removed. To ensure credibility and validity, before being anonymised, each participant had the right to reread these answers/statements and was free to request that they be removed (or not). However, no complaints were registered in the sense of deletion. We then entered and stored these data in an NVivo database for qualitative analyses.

All quantitative data collected were aggregated and anonymised. Outliers were processed during quality check in Microsoft Excel software 16.0, developed by Microsoft in Redmond, Washington DC, USA. The data were then exported to IBM SPSS (Statistical Package for Social Sciences) software 20 for statistical analysis.

Throughout all these data management procedures, the ethical considerations of the research were strictly respected. It should be noted that this study is part of a larger doctoral research project whose protocol was initially approved by both the Institutional Review Board of ITM Antwerp (IRB/RR/AC/187/1468/21) and the Medical Ethics Committee of the University of Lubumbashi (UNILU/CEM/034/2021). Participation was free and voluntary for all participants. Before participating in the FGDs or the interview, each participant provided free and informed consent orally.

We did not register any cases of refusal or withdrawal. We made a commitment and ensured that the information provided remained strictly confidential, even though it was not really sensitive given that the patient data had been aggregated and anonymised beforehand.

### 2.7. Data Analysis

Given the nature of the data collected, two types of analysis were performed: descriptive statistical analysis for quantitative data and content analysis for qualitative data. Statistical analysis enabled us to describe the health outcomes of integration by measuring several indicators. First, we measured the accessibility of mental health services using access indicators (including the availability of human resources and psychotropic drugs, geographical accessibility, financial accessibility, etc.). Second, we measured the use of these services by calculating indicators related to the functioning of health services, including mental health services. Third, the clinical profile of health service users was described by calculating the proportion of cases that requested this type of service.

In carrying out the content analysis, we (i) explored stakeholders’ perceptions of the programme regarding the reduction (or otherwise) of mental health-related stigma; and we (ii) explored and synthesised the lessons learned from these two integration experiences, testing the mhGAP. We then analysed the strengths, weaknesses, opportunities and threats, at the end of which we identified the challenges of implementing integration and solutions to achieve sustainability of these experiences.

## 3. Results

### 3.1. Characteristics of the Study Participants

A total of 37 participants (met face-to-face and online), of whom 76% were men, participated in the study. Of the participants, 34 were from the DRC (Lubumbashi, Matadi, Kinshasa, and Bukavu), 2 from Belgium (Antwerp), and 1 from Guinea (Conakry). Among them, 51% came from the Ministry of Public Health, 27% from academia, and 22% from health non-governmental organisations (NGOs). A total of 42% of the participants were healthcare providers, 27% were public health experts, 14% were mental health experts, 14% were NGO stakeholders responsible for implementing integration programmes, and 3% were public health policymakers. The experts from Belgium and Guinea are doctors (*n* = 2) and sociologists (*n* = 1), all with a PhD in public health and very long experience in the field of public mental health. They have been involved both in the integration of mental health into primary care in Guinea and Belgium and as external advisers in one of the two Congolese integration experiences we studied.

### 3.2. Description of Interventions Planned and Implemented in Programmes U and R

Various interventions (i.e., one-off and cross-cutting activities) recommended by the mhGAP-IG were planned and implemented as part of integration programmes U and R in the two health districts. Table 3 summarises the activities planned and implemented in the two programmes.

In both programmes, primary care providers (non-specialists in mental health) and other community stakeholders (non-health professionals) were trained in the modules defined in the mhGAP-IG and in other essential concepts on mental health and the integration of services. Table 4 presents the content of the training received by primary care providers and other community stakeholders.

Following capacity building, seven priority MNS disorders contained in the mhGAP-IG have been included in the package of integrated mental health services in primary care facilities of two programmes. These include depression, psychoses, epilepsy, dementia, mental and behavioural disorders in children and adolescents, substance use disorders, and self-harm and suicidal behaviour. In addition, given the health emergency context (COVID-19 pandemic) in which Programme U was launched, anxiety and stress disorders were also considered as separate components of other significant mental health complaints.

At the end of the training, some mental health tasks were delegated to primary care providers and community health workers working in the district hospital, health centres and the community (Table 5). A small mental health ward was established at Mangembo district hospital, with a capacity of around ten beds, for the short-term hospitalisation of patients with disruptive behaviour considered to be psychiatric emergencies that were referred by the peripheral health centres. This delegation of mental health tasks from specialists to primary care providers was consolidated by formative supervision, mentoring, and support activities, during which difficult cases were initially managed with the support of specialist coaches. Training and supervision were organised as an interactive and participative learning trajectory through practise.

After the task shifting, the primary care facilities were supplied with essential (psychotropic) drugs. A total of 11 drugs were available at PHC facility in programme U et 5 among these in programme R. For programme R, additional drugs were available only in district hospital. It should be noted that the essential drugs supplied were those recommended in the mhGAP-IG.

### 3.3. Health Outcomes of Integration

#### 3.3.1. Accessibility of Health Services

Data analysis showed that integrating a mental health service package into PHC systems helps improve accessibility to mental health services. In terms of the availability of services, the analysis revealed that the two health districts currently have healthcare provider teams and community health workers trained in mental health. Table 6 presents the professional profile of the available health staff and other stakeholders trained.

With regard to the availability of key health personnel (doctors, nurses, and midwives), data analysis showed that in programme U, 22 health professionals working at the health centre of Tshamilemba had been trained in MHPSS, including the use of mhGAP-IG. More specifically, 12 of the 17 nurses, 8 of the 15 doctors, and 2 midwives were trained. In programme U, one doctor—the mental health referent—received practical training in neuropsychiatry at the Sendwe provincial general referral hospital, and four community health workers available in the Agétraf health area were briefed on mental health. In addition to these primary care providers, two clinical psychologists (1.7 clinical psychologists per 10,000 inhabitants) and one social worker (0.8 social workers per 10,000 inhabitants) were recruited and trained in the use of the non-pharmacological interventions recommended in the mhGAP-IG. In programme R, 5 of the 10 physicians and 17 of the 147 nurses currently working in health facilities (i.e., 0.3 health professionals per 1000 inhabitants in Mangembo district) were trained in MHPSS, including the mhGAP-IG. Furthermore, as part of this programme, a nurse—who acts as a mental health referent at the district hospital—was trained in mental health at the Mont Amba Neuro-Psycho-Pathological Centre, and four nurses were sent for further internship to the Telema mental health centre in Kinshasa.

With human resources in place whose capacities were strengthened and medicines made available (as described above) in healthcare facilities, the provision of mental health services was launched. In Tshamilemba, mental health provision was made accessible in a single health centre of Tshamilemba in one of the district’s thirteen health areas, namely the Agétraf health area.

In Mangembo, mental health services were offered in 19 healthcare facilities spread across six of the district’s ten health areas. Note that this is a rural health district covering an area of 2153 km^2^, which caters for the health needs. Regarding the situation presented, one participant stated the following:


*“… it is estimated that to date more than 50% of the population still travels a distance of more than 5 km to access mental health services in Mangembo”. (P10, FGD3, PU)*


An analysis of the availability of essential medicines and other products revealed that in the district using programme R, 47.4% of the 19 healthcare facilities had psychotropic medicines in stock, while only 1.7% of the 59 healthcare facilities in the district using programme U. In both programmes, and especially in programme U, occasional stock-outs of one or more drugs ranging from 2 to 10 days were reported. In a programme, a participant in the FGD stated that he occasionally resorted to informal drug supply. He said:


*“We follow the normative drug supply circuit… But, with occasional recourse to approved private suppliers”. (P8, FGD3, PR)*


There have been shortages of essential psychotropic medicines, particularly in programme U, due to a halt in funding.

The affordability of mental healthcare and services provided in healthcare facilities was analysed. In programme U, mental healthcare and services were fully paid for out-of-pocket, including medication, because the programme did not introduce flat-rate pricing. The cost of mental healthcare and services exceeded USD 20 for an episode of mental disorder. In programme R, flat-rate pricing was applied. At the health centre, the flat rate was approximately CDF 15,000 (equivalent to USD 6) for an episode of mental disorder; CDF 23,500 (equivalent to USD 9) for outpatient care; and CDF 75,000 (equivalent to USD 28) for inpatient care.

#### 3.3.2. Use of Healthcare and Services

The analysis of the use of curative services, including for MNS disorders, in primary care settings showed an improvement in (mental) health service use indicators in both intervention areas since the integration of a mental health service package into the PHC system. In the district using programme U, the general curative consultation rate increased from 38 NC/100 inhabitants/year in 2022 to 42 NC/100 inhabitants/year in 2024; while the curative consultation rate for MNS disorders, which was unknown at the start of the programme, reached 7.1 NC/1000 inhabitants/year in 2022, then 14.4 NC/1000 inhabitants/year in 2024. On the other hand, in the district using programme R, the general curative consultations rate in health centres increased from 40 NC/100 inhabitants/year in 2023 to 43 NC/100 inhabitants/year in 2024, while the curative consultation rate for MNS disorders, which was unknown at the start of the programme, reached 9.4 NC/1000 inhabitants/year in 2023, then 14.2 NC/1000 inhabitants/year in 2024 (Table 7).

Estimation of mental healthcare utilisation indicators at district hospitals using programme R revealed an increase in hospitalisation rates for MNS disorders. This hospitalisation rate, although low overall, doubled over the study period, increasing from 1.3 per 1000 inhabitants in 2023 to 2.5 per 1000 inhabitants in 2024 (Table 8).

The participants in the FGDs indicated that the mental health services available in their healthcare facilities were considered acceptable by the various stakeholders, particularly the service users, as they contributed to improving the quality of services. The fact that healthcare providers collaborate with traditional and spiritual healers strengthens users’ trust in “Western” healthcare settings.


*“Personally, I decided to go to XX health centre for mental health care because my pastor referred me there. He reassured me that there was no incompatibility between prayers and psychiatric care… I find that this collaboration between nurses, doctors and my pastor strengthen trust in mental health care”. (P5, FGD4, PU)*


The average occupancy rate of beds for mental health problems in the hospital increased from 41.3% in 2023 to 80.4% in 2024. The average length of hospital stays for psychiatric patients increased from 12 days in 2023 to 14 days in 2024.

Data analysis in relation to the functionality of the referral and counter-referral system revealed that the rate of mental health referrals increased in one of the health districts. The rate increased from 0% at the start of the programme to 12.9% in 2024 in the district using programme U, while the counter-referral rate increased from 0% at the start of the programme to 3.5% in 2024. On the other hand, in the district using programme R, although the participants in the FGDs stated that the counter-referral system was functional, referral, and counter-referral rates were not reported.

Data analysis showed that of all the people who received mental health services in the healthcare facilities integrating the mental health provision, in the district using programme U, 27.9% received services in primary care settings for other significant mental health complaints, including anxiety and stress disorders, followed by 26.4% who suffered from moderate to severe depression, and 17.7% due to epilepsy. In contrast, in the district using programme R, the majority of people cared for in primary care settings suffered from depression (42.8%), epilepsy (23.3%), and psychosis (20.7%); while those cared for in the district hospital suffered from moderate to severe depression (35%), psychosis (24.9%), and epilepsy (24.9%) (Table 9).

Of the people treated under programme U, 100% received psychological interventions (such as psycho-education and counselling to all patients, or interpersonal therapy, cognitive-behavioural therapy, etc., to others, depending on their mental condition), 84.7% received pharmacological interventions, and 55.3% received social interventions, in particular, psychosocial support. Of the individuals taken into care for mental health reasons, the partial or total remission rate reported by programme U was 74.1%, and the drop-out rate was 5.9%, while in programme R it was, respectively, 70.8% and 3.2%.

#### 3.3.3. Stakeholders’ Perceptions of Mental Health-Related Stigma Reduction

In Programme R, where community awareness-raising activities were launched, stakeholders observed a partial and positive change about the stigmatisation and mistreatment of people with severe mental disorders. The KIs involved in the FGDs stated that the social distance between the population and those with mental health problems was decreasing. One of them put it this way:


*“Now people are not very afraid of people with mental illnesses. They accept that it is normal for someone to get agitated and to talk too much. Those close to them are now prepared to take them to the health centre for mental care”. (P9, FGD2, PR)*


The reduction in mental health-related stigma and social rejection seems perceptible in the increasingly rare presence of people suffering from severe mental disorders on the street in the health district using programme R. One participant stated that these people with mental illness are no longer rejected by society, stigmatised, or discriminated against, as was previously the case. In particular, he said:


*“There is no longer a multiplicity of cases of wandering of mentally ill people in the area, nor many human rights violations such as stigmatisation, discrimination, physical abuse, abandonment and social rejection, which were observed before…”. (P3, FGD1, PR)*



*“The region’s mentally ill people now receive care in health centres without encountering the same difficulties as before… We are delighted”. (P6, FGD2, PR)*


A healthcare user recognises the progress made in respecting the rights of people with mental health problems in healthcare facilities that have integrated mental health provision. According to him, primary care providers in programme U increasingly respect patients’ rights and are mistreating them less and less. He stated:


*“We increasingly can overcome the fear of stigma associated with our mental health condition, because the centre meets our needs… We are no longer blamed as we used to be and professional secrecy is now respected”. (P11, FGD2, PU)*


In the district using programme U, stigmatising attitudes towards people with mental health problems may still be perceived at the community level. A participant in the FGDs stated:


*“There are still people here and there who believe that the mentally ill are dangerous. However, in reality, mentalities are changing, and stigmatising attitudes are becoming less and less perceptible. If we organise awareness-raising sessions, we can significantly reduce the social stigma associated with these patients”. (P5, FGD3, PU)*


These stigmatising attitudes appear to be linked to the almost total absence of mental health awareness and education activities. A participant in the FGDs stated:


*“There seems to be a partial change in stigma… However, this change in stigmatising attitudes, practises and reactions and other unfavourable behaviour in the community towards mental health remains a challenge. We should redouble our efforts in this area by raising awareness”. (P7, FGD2, PU)*


### 3.4. SWOT (Strengths, Weaknesses, Opportunities, and Threats) Analysis of the Two Integration Programmes Testing the Proposed mhGAP

Our analysis of integration programmes testing the proposed mhGAP revealed some strengths and opportunities, as well as weaknesses and threats that are challenges for (local) health systems. The SWOT analysis is summarised in Table 10.

### 3.5. Lessons Learnt from Experiences Testing the WHO’s mhGAP

Several lessons have been learned from these two Congolese integration experiences. They are listed below according to the three phases of the mhGAP-IG.

#### 3.5.1. Lessons Learned in Relation to Assessment and Planning

It is essential to identify all key stakeholders (i.e., stakeholders from the health, social, education, and nutrition sectors, etc.) at the outset and to clarify the institutional arrangements between the parties involved in the integration process. This would avoid conflicts of competences and interests that could lead to the failure of integration experiences.Humanitarian crises (whether health-related or man-made) can open a policy window for advocating and implementing the integration of mental health into PHC settings. This was the case in Tshamilemba district, where integration was launched during the COVID-19 pandemic context. In such cases, it is important to sustain the integration achievements launched in an emergency context.Mental health integration activities designed using a participatory approach (with the health authorities and the support of the mental health programme) and included in the development and operational action plans of health districts are more likely to be easily implemented and sustainable. This alignment with the health district development plan facilitates the organisation of joint and integrated supervision as well as monitoring of SNIS indicators. Empowering the health district management team increases the chances of success, especially when it is supported by a range of local, provincial, and national specialists.Integration using domestic financial resources should be prioritised. To this end, the creation of a budget line dedicated to integration activities in the State budget allocated to the health sector could be an avenue to pursue. When integration is launched with financial support from international partners, it is important to carefully consider the exit strategy. Technical and financial partners invest in the integration of mental health, either as part of the development of the health district or as part of emergency projects. Their programmes are therefore both temporal and spatial. When planning to intervene in a district or region, a disengagement plan must be discussed and adopted at the start of the intervention and then respected by all parties involved. Care must be taken to ensure that the State party takes ownership of the process and can continue integration efforts after the end of external funding.

#### 3.5.2. Lessons Learned Regarding Preparedness

Training healthcare personnel (primary care providers, community health workers, etc.) involved in integration is essential, but is not enough on its own. It is important to organise support, mentoring, and supervision activities with the support of specialists, the district management team and peers. A combination of these strategies is essential for anchoring mental health literacy and clinical skills.Integrating mental health services into poorly functioning or non-functioning health facilities and/or (local) health systems may result in low utilisation of the health services provided. Before or during the integration process, it is preferable to improve the functionality of healthcare facilities to strengthen the healthcare system.Traditional and spiritual healers are an important category of actors in the process of integrating mental health. They play both the role of healthcare providers (traditional and spiritual) and community connectors. Their involvement in the process, particularly in the mental health platform, would have a positive impact on the acceptance of the mental health services integration strategy put in place. This approach would promote collaboration between traditional, alternative, and psychiatric care providers and improve the perception of mental health problems.The support provided to the local “social” sector (i.e., formal and informal community organisations) by healthcare providers significantly contributes to the promotion of community-based mental healthcare. This will enable them to ensure the ongoing awareness, mobilisation, and commitment of leaders and community members. This guidance makes it possible to identify individuals experiencing common MNS and referral them to primary care facilities.Harmonising the consultation sheet by automatically including the mental health component and recruiting mental health staff, such as clinical psychologists, mental health nurses and psychiatric social workers, in primary and secondary care facilities is an efficient measure. This would make it possible to reduce the costs associated with the mobility and motivation of specialists invited to mentor and supervise primary care provider teams.

#### 3.5.3. Lessons Learned Regarding Provision of Services

In the Congolese emergency and development context, it is feasible to effectively integrate a mental health services package into PHC systems and to provide care for common MNS disorders in primary care settings, as well as in the community.It is important to follow the integration process that uses the WHO’s mhGAP-IG while adapting it to the Congolese context. It is then necessary to normalise the process by establishing harmonised and/or rationalised national integration guidelines.Most people with common MNS disorders are treated/cared for in frontline healthcare facilities on an outpatient basis. There is a need to reinforce this line of healthcare at the community level, which provides traditional and informal care as well as follow-up care.Maintaining reform movements in the initial training of physicians, nurses, and mental health technicians requires an internship in a psychiatric ward within a psychiatric hospital. It is preferable, as far as possible, to increase the number of hours of theoretical and practical training and to prepare clinicians for the role of training supervisor. The mhGAP-IG modules should be integrated into the initial training curricula of future primary care providers.The availability of psychotropic drugs, supported by the purchase and renewal of stocks, and performance-based motivation and tasks of the district health management team and healthcare providers at different levels of the care system pyramid contribute to improving the quality of services. In this sense, tiered care (second- and first-line healthcare facilities, as well as the community) by motivated staff, based on the mhGAP-IG adapted to the local context, can considerably improve complaints and facilitate the reintegration of people who have benefited from care into their usual living environment.Integrating mental health services into the PHC system improves the overall use of health services, particularly curative services, in primary care settings. This may be due to the humanisation of care resulting from changes in the attitudes of healthcare providers.The acceptability of a new range of health services, specifically mental health services, as well as changes in the population’s behaviour regarding their use and the mental health-related stigma, are long-term processes. Indeed, despite the enthusiasm shown during the awareness-raising sessions, the stakeholders involved in the integration experiences noted that the population needs more time to assimilate the new knowledge and practises transmitted.Integration of mental health education may be technically successful, but the use of services is low because of a lack of funding. In LICs such as the DRC, most mental health service users belong to the indigent class. If the cost of care remains catastrophic and impoverished, healthcare facilities that provide mental health service are likely to desert.The involvement of clinical psychologists and social workers alongside psychiatrists to support non-specialist care providers in primary care settings is beneficial. In this sense, the integration of mental health professionals, such as clinical psychologists and social workers, into the primary care provider team may be necessary to provide holistic care and promote the sustainability of interventions after external financial support has ceased.There is a need to move from an isolated project approach to an integrated services programme approach, with the aim of strengthening (local) health systems by improving the functionality of healthcare facilities. A retention policy should be implemented for non-specialist care providers benefiting from capacity-building programmes for the implementation of the mhGAP-IG. This would make it possible to avoid repeating training cycles for primary care providers instead of organising refresher sessions.

## 4. Discussion

The results of this study, which aimed to document the health outcomes of Congolese integration experiences, testing the feasibility of the WHO’s mhGAP and to draw lessons learned from these experiences, show that it is feasible and effective to integrate a mental health services package into primary care settings, in both unstable emergency and more stable development contexts in the DRC. This study highlights encouraging findings despite the implementation challenges that (local) healthcare systems still need to address.

Nearly 2000 people were treated for mental health disorders in health centres at the district hospital, under both programmes, from 2021/22 to 2024. Between 70% and 75% of those treated recovered. In view of these results, we can readily admit that the integration testing of the WHO’s mhGAP in the Congolese context is feasible and effective and suggest that the interventions as described were largely consistent with the ToC adopted [28]. During the study period, the curative consultation rate for MNS disorders, which was unknown at the start of the programme U, reached 7.1 NC/1000 inhabitants/year in 2022, then 14.4 NC/1000 inhabitants/year in 2024 in Tshamilemba. This curative consultation rate for MNS disorders, which was also unknown at the start of the programme R, reached 9.4 NC/1000 inhabitants/year in 2023, then 14.2 NC/1000 inhabitants/year in 2024 in Mangembo district. Although these rates of use are relatively low, these results support the postulate that in the Congolese urban and rural context, it is possible to integrate mental health into primary care settings, provided that the necessary resources are allocated [29,30] and that there is a firm commitment from the stakeholders in the healthcare system [31,32].

The majority of people who have received care for mental health reasons have done so on the front line and on an outpatient basis. Our professional experience in the DRC shows that people prefer to consult health centres because they know that they will return home the same day, unless the illness for which they are consulting is serious. One of the objectives of integration is to promote the provision of primary mental healthcare in an open setting [33]. This may argue in favour of therapeutic interventions organised in an open setting, i.e., in health centres and at home or in the community.

The data used to calculate the indicators listed above come from two sources considered to be weak: (i) the healthcare facilities that generated them themselves, such as the health centre of Tshamilemba; and (ii) data from the national routine health information system that reports data after a validation process within the health district management team. It should be noted that the statistical data used were generated by healthcare providers who received no special payment for this task and whose remuneration conditions were often contested. In the absence of substantial remuneration, they could therefore afford to neglect their duties and be biased in their reporting. However, the quality of the data was enhanced by the quality of the discussions with the various stakeholders in the FGDs, including public (mental) health experts and academics who, as noted, were independent-minded. We also point out that the figures used in the denominator to calculate indicators are generally based on estimates. This calls for caution when interpreting these indicators.

The overall rates of use of curative consultations at primary care facilities in the two health districts for mental health reasons remained low. Low utilisation of mental health services remains a challenge in many countries [34]. It is very likely that financial factors negatively influenced the use of curative care. In other programmes, such as those providing mental healthcare and psychosocial support to survivors of sexual and gender-based violence, where services were free at the point of use, the utilisation rate was estimated at 9 per 100 inhabitants per year [5]. In addition to financial constraints, the belief in supernatural causes of mental health disorders, which is widespread in LMICs, has been identified as one of the factors explaining the low use of mental health services [35,36], particularly in contexts of exclusive health systems that do not allow any collaboration between Western, traditional, and alternative medicine. Furthermore, an intra-provincial or even inter-provincial comparison of these indicators remains questionable because of the pilot (and thus unrepresentative) nature of the integration experiences. Moreover, until recently, the official Congolese health information system used other indicators and definitions of mental health disorders (psychosis, neurosis, and epilepsy), which were quite different from those used in our studies [11].

Even if the integration of mental health is sometimes advocated by certain healthcare providers, carers, and managers, it does remain problematic for other primary care providers [34,37,38]. Fear of work overload, fear of being labelled as the carers of the “crazies”, difficult working conditions, and unsatisfactory salary conditions could be some of the reasons explaining their reluctance.

Research has also shown that some people with mild mental health problems do not see the need for seeking help because they do not properly recognise their symptoms [39,40]. Others with mental health problems refuse to go into care because they believe that health professionals may not take them seriously. They are oftentimes stigmatised or discriminated against when they attend healthcare facilities providing mental healthcare (including general healthcare facilities) [41,42].

The results indicated that the rate of general curative consultation gradually increased, rising from 38 NC/100 inhabitants/year in 2022 to 42 NC/100 inhabitants/year in 2024 in Tshamilemba and from 40 NC/100 inhabitants/year in 2023 to 43 NC/100 inhabitants/year in 2024 in Mangembo. Such an increase, although not drastic, is satisfying because people who had (and still have) unmet (mental) health needs can then turn to primary care facilities with the hope of receiving adequate care, thus increasing the likelihood of (local) health systems improving their progress towards UHC. The integration of mental health services contributes to an increase in curative consultation for “walk-in” patients in general health facilities [43]. This seems to be partly due to the integration of mental health, among other things, to improving the quality of care offered to patients in a person-centred care approach [43,44]. Primary care providers report high partial or total remission rates (74%), but without providing clear objective criteria for this short-term remission. Conducting surveys involving patients undergoing treatment to obtain long-term follow-up data for these patients would make it possible to confirm (or not) these reported remission and/or recovery rates.

Of the people who used mental health services, the majority suffered from anxiety disorders (27.9%), depression (26.4%), and epilepsy (17.7%) in programme U; while the majority suffered from depression (35%), epilepsy (24.9%), and psychosis (24.9%) in programme R. In cases of depression, diagnosis is often made without distinction regarding the degree or type of disorder. The prevalence of depression is significantly higher than that found in general population studies published in a systematic review with meta-analysis of publications from five continents, estimated to be 18% [45]. Could there be a diagnostic problem? It is possible that the tools used to diagnose mental health disorders are based on a categorical approach. It is not a scientifically correct to always classify people with similar symptoms but facing very different social, cultural, and/or economic situations (poverty or other financial problems, unemployment, interpersonal difficulties, and family violence, structural violence, community conflicts, migration, etc.) as presenting a single specific health problem [46]. This counting of symptoms, without taking into account context and diagnostic heterogeneity [47], may be at the root of the overestimation of cases of depression and perhaps other mental health disorders in the Congolese context. The same applies to the prescription of psychotropic drugs using protocolised solutions [5]. Three out of ten people were affected by anxiety disorders in the urban environment of Lubumbashi. This may be explained by the fact that integration was launched during the COVID-19 pandemic. In addition, the current lifestyle in both urban and rural areas of the DRC is likely to influence the increase in other significant mental health complaints, including stress and anxiety disorders. As a single pathological entity, they receive little attention in primary care settings. Would it not be appropriate to separately consider these other significant mental health complaints (i.e., anxiety disorders and stress disorders) so that they receive the attention they deserve during the mental health integration process? The high prevalence of psychoses in hospitals can be explained by the fact that these mental health disorders are the most prevalent in our context. Generally speaking, if financial resources are available, families and friends rush their loved ones to the hospital as soon as they realise that they are beginning to show signs of psychotic behaviour.

Of the patients treated in primary care facilities in the Tshamilemba health district, 100% received psychological interventions, 84.7% received pharmacological interventions, and 55.3% received social interventions (guidance, social support, accommodation, food, family mediation, etc.). These results support recruiting clinical psychologists at the first line of care. It is advisable that MNS disorders such as acute stress, intellectual disability, and mental and behavioural disorders in children and adolescents should not be the subject of medical prescriptions in primary care settings. Providers of progressive psychological interventions would therefore be invaluable. It has also been found that almost nine out of ten people are treated with medication. Is this due to over- or irrational prescription of psychotropic drugs? In a previous study [5], we also observed that the psychiatric aspects of healthcare were given priority to the detriment of the psychological, sociocultural, and spiritual aspects of caring for people with mental health problems. Over-medicalisation risks have a considerable negative effect on the well-being of people who apparently require other investments from providers. In our experience of accompanying teams of primary care providers in the process of integrating mental health, we have often found that these health professionals sometimes find it difficult to look beyond what is recommended in management protocols. Furthermore, is it possible that over-prescription of drugs is linked to the commercialisation of mental healthcare in a context of shortage of psychotropic drugs? [47].

Our results demonstrate that integrating mental health improves the functionality of the referral and counter-referral in health districts. From 2022 to 2024, the mental health referral rate in one health district increased from 0% at the start of the programme to 12.9% in 2024, while the counter-referral rate increased from 0% at the start of the programme to 3.5% in 2024. These results suggest that primary care providers are gradually recognising the importance of the continuum of care in the management of mental health problems [48].

The results revealed that access to mental health services in the two health districts was gradually improving. At the health centre of Tshamilemba, in the Tshamilemba health district, 68.8% (22/32) were trained in mental health, including the use of the mhGAP-IG; while in the Mangembo health district, 8.9% (22/247) were trained. Note that in the Mangembo district, the programme is still in its second year out of the five planned. In Tshamilemba district, 1.8% (only 1 out of 55!) of the health facilities are supplied with essential medicines, compared with 47.4% (9/19) of health facilities in Mangembo. While these results are encouraging, major efforts are needed to cover all health districts. Well-trained human resources for health and the availability of quality medicines are two important building blocks of the (local) health system that are likely to support efforts to integrate mental health [47]. It is known that healthcare facilities based in these local health systems cover about 80% of the population’s health needs [13]. One of the best strategies for ensuring the availability of essential psychotropic medicines is to set up a drug credit line to support the primary care facilities that need them most.

Despite the efforts made in the two health districts in which the integration of mental health is being tested, the stigma attached to mental health remains strong. While some participants in the FGDs felt that people were no longer very afraid of people suffering from mental health problems, that they accepted that it was normal to develop these problems, and that they were prepared to take them to a healthcare establishment, others, on the other hand, said that people still believed that these patients were dangerous and that they could still reject them, calling for more awareness-raising initiatives. Indeed, the fear of social stigmatisation is still often reported by users of mental health services, all the more so when they attend “Western” healthcare facilities [5]. This stigma remains one of the major obstacles to access to mental health services and is thought to be at the root of mistreatment and rejection of patients by those close to them (family, friends, employers, etc.) [49]. It turns out that this social and/or structural stigma becomes even more problematic when it emanates from the providers themselves within health facilities [50]. To significantly reduce mental health-related stigma, investing in community-based mental healthcare approaches is important.

Several lessons can be learned from these Congolese experiences of mental health integration that tested the WHO’s mhGAP, and some solutions have been proposed to promote its sustainability. However, to ensure greater sustainability, (local) health systems need to take on board the lessons learned from these experiences. The lessons appear to corroborate those of other experiences conducted in LMICs that tested the mhGAP [51,52]. These studies have learned that it is necessary to collaborate with local stakeholders. It is necessary to build on PHC systems to reach the most vulnerable groups of people with unmet needs. To identify people with (potential) mental health problems, it is important to use locally understandable concepts. Primary care providers must be adequately remunerated for the healthcare (including mental healthcare) they provide. It is appropriate to take advantage of the opportunities that arise in specific crisis situations to achieve integration, as was demonstrated in the case of Tshamilemba health centre facing COVID-19. However, care must be taken to ensure that integration measures introduced at the time of these crises are sustained.

The ambitious nature of the two programmes, which aimed to treat the seven common MNS disorders and provide up to 11 types of medication, but which were not commensurate with the financial and logistical resources deployed, would have led to minimal outcomes. We therefore suggest being very pragmatic in the choice of priorities when an integration programme is being considered and, for the two programmes currently underway, relegating certain MNS disorders that are infrequent in the context to the background, to devote the necessary energy to the disorders that are really a priority.

### 4.1. Study Strengths, Limitations, and Implications

This study has three main strengths. The first is the type of design used. Indeed, the multiple case study design is recognised as one of the most powerful in terms of generating solid evidence in the absence of quasi-experimental study designs [25]. However, as multiple case study designs do not allow causal links to be measured, further research would ideally adopt more robust prospective designs with specific outcomes of interest being measured across different stakeholders. The second lies in the fact that it documents, for the first time to our knowledge, two experiences of “formal” integration of mental health services purposely guided by WHO’s mhGAP. Accordingly, these results stand a better chance of being comparable with those of studies carried out in similar contexts. Third, unlike other studies that only used secondary or primary data, this study used both primary and secondary data from different collection methods and sources. This triangulation allowed us to minimise information bias and thus improve the quality of the provided information.

However, there are two main limitations. First, with the available data, it was not possible to analyse the situation by distinct geographical health areas and establish where the patients using the Tshamilemba health centre actually come from. The same holds true for the Mangembo health district for programme R. Overall, the rates of use of curative mental health services remain low. It would have been worthwhile to monitor this utilisation over longer periods of time than was the case in the present study. The mhGAP tools used for analysing integration policies do, however, not address the reality of traditional healers providing mental healthcare. Given the fact that these traditional care providers are, and will, remain in the medium term an important source of mental healthcare for people (whatever the effectiveness of this source of care), it is justified to take them into account in future integration experiences. It may therefore be indicated to enrich the current mhGAP framework in that respect.

The results of this study raise several other issues that merit further consideration. First, it is important to thoroughly document the experiences so as to be able, eventually, to translate the findings into health policies. In a perspective of scaling up mental health services, there is need to map all the PHC facilities currently offering mental care, assess the level of integration, certify those that have reached the required level, and then raise the level of those in the process of integration. From there, a roadmap for the progressive integration of mental health can be established. Finally, it is necessary to introduce mhGAP-IG modules into the initial training curriculum for future primary care providers.

### 4.2. Reflexivity

During this study, one of researchers (EMM) was not entirely neutral. He was indeed involved in the action research process conducted at the health centre of Tshamilemba, as an external researcher. In addition, in his capacity as a senior manager at the DRC’s PNSM, he was also occasionally invited to take part in reflection sessions on the progress of the programme implemented at Mangembo (in programme R). These combined roles may have been a source of bias.

## 5. Conclusions

The burden of mental health disorders is substantial in the DRC, and innovative experiences to fill the huge gaps in treatment and care are urgently needed. Mental health integration experiences testing the mhGAP, based on a strategy of delegating mental health tasks to primary care providers, have been (and are currently being) conducted. Although the full results of the impact of these experiences are not yet available, the intermediate health outcomes obtained indicate that it is possible to improve access to and use of mental health services in the DRC by using the WHO’s mhGAP as an integration guidance. This guideline is pragmatic, cost effective, and promising in the Congolese context. For the moment, everything seems working well to the satisfaction of the mhGAP operations team. However, we remain cautious about the sustainability of the longer-term impacts of these programmes once external funding has ended. It would be useful to examine whether these achievements will be maintained in the longer term during the planned capitalisation activities and final evaluation. In addition, concerted action by a range of health system stakeholders can make it possible to meet the identified challenges and make this integration guideline effective. Weak institutional ownership of integration experiences developed in these two health districts remains however a challenge, which needs to be addressed too.

## Figures and Tables

**Table 1 ijerph-22-00457-t001:** Characteristics of the Tshamilemba and Mangembo districts at the start of the programmes.

Characteristics	Tshamilemba District	Mangembo District
Surface area	42 km^2^	2153 km^2^
Number of health areas (or sub-districts)	13	10
Population ^#^	283,000	80,417
Number of healthcare facilities ^#^ (% of primary care facilities including health centres and district hospital)	59 (93.2%)	31 (90.3%)
Number of registered healthcare providers ^#^ (% of primary care providers)	338 (75.1%)	240 (85.4%)
Rate of mental health professionals (excluding traditional mental care providers) per 100,000 inhabitants	Unknown	Unknown
Curative healthcare utilisation rate (NCs/inhab/year) ^#^	0.38	0.40
Rate of use of curative mental health services, in general, curative consultations (NCs/inhab/year) ^§^	0	0
Mental health service coverage rate	Unknown	Unknown

^#^ Data source: Tshamilemba and Mangembo district operational action plans; provincial health development plans; national health development plan; NCs stands for new cases; ^§^ Based on our calculation. Source: [3,6,7,8,12].

**Table 2 ijerph-22-00457-t002:** Characteristics of the cases.

Case	Start	Status	Funding	Localisation	Scale of Implementation	Nb Integrated Facilities (Nb Beneficiaries)	Intervention Type
Programme U	2021	Ongoing	ITM/DGD	Tshamilemba	Health area	1 (12,000 inhabitants)	Action Research
Programme R	2022	Ongoing	Memisa/DGD	Mangembo	Health district	19 (80,417 inhabitants)	Development project

ITM: Institute of Tropical Medicine in Antwerp, DGD: Belgian Directorate-General for Development Cooperation and Humanitarian Aid.

**Table 3 ijerph-22-00457-t003:** Description of activities planned and implemented under Programmes U and R.

Interventions (Activities) Recommended for Integrating Mental Health into the PHC System	Programme U Interventions		Programme R Interventions		Remarks
	Planned	Implemented	Planned	Implemented	
Phase I: Plan					
1. Setting up an mhGAP operational team	✓	✓	n/d	✓	Programme U had an action research team, whereas for programme R there are no data available on the mhGAP team.
2. Conducting a situational analysis	✓	✓	✓	✓	
3. Developing an mhGAP operational plan and budget	✓	✓	✓	✓	Funding for programme U was halted in 2023.
4. Advocating for mental health	✓	✓	✓	✓	
Phase II: Prepare					
1. Adapting the components of the mhGAP package	✓	-	✓	-	The mhGAP-IG modules were not adapted to the context due to time and resource constraints.
2. Training staff in mhGAP	✓	✓	✓	✓	
3. Preparing for clinical and administrative supervision	✓	✓	✓	✓	
4. Coordinating care pathways	✓	✓	✓	✓	
5. Improving access to psychotropic drugs and improving access to psychological interventions	✓	✓	✓	✓	Implementation of programme U was made possible by recruiting clinical psychologists and social workers.
Phase III: Provide					
1. Providing services at facility level	✓	✓	✓	✓	
2. Providing treatment and care in the community	✓	✓	✓	✓	
3. Raising awareness of MNS conditions and available services	✓	-	✓	✓	In Programme R, the activity was still in its early stages.
4. Supporting the implementation of prevention and promotion programmes	✓	-	✓	-	
5. Including mental health data into the SNIS	n/d	-	✓	-	The data were not yet included in the DHIS2 mental health module.
6. Creating demand for mental health services	✓	-	✓	✓	The intervention was not carried out in programme U and only at an early stage in programme R.

mhGAP-IG: mental health Gap Action Programme-Intervention guide; SNIS: National Health Information System; n/d: no data. Data sources: document review, focus group discussions, and interviews; ✓: activity carried out; -: activity not carried out.

**Table 4 ijerph-22-00457-t004:** Description of the content of staff training courses.

Main Themes (Sessions) Developed During the Course	Programme U	Programme R
Content for healthcare providers		
General information on the MHPSS ^1^ and the integration of mental health services	✓	✓
Investing in mental health	✓	
Improving access to care to address MNS disorders	✓	
Introduction to mhGAP-IG and the general principles of care	✓	✓
Depression	✓	✓
Psychoses	✓	✓
Epilepsy	✓	✓
Dementia	✓	✓
Child and Adolescent Mental and Behavioural Disorders (CMH)	✓	✓
Substance use disorders	✓	✓
Self-harm and suicidal behaviour)	✓	✓
Other significant mental health complaints	✓	✓
Mental health information management	✓	✓
Content for community health workers		
Basics of mental health, mental health problems, and care	✓	✓
Case identification		✓
Community-based management techniques		✓
Communication for social and behavioural change	✓	✓
Home visits, patient referral, and reporting		✓

^1^ MHPSS: mental health and psychosocial support; ✓: session developed during the course.

**Table 5 ijerph-22-00457-t005:** Mental health service package integrated into PHC systems in Tshamilemba and in Mangembo.

Mental Health Service Package	District Hospital Level	Health Centre Level	Community Level
Curative services			
Case identification and referral	✓	✓	✓
Curative medical consultation	✓	✓	
Curative psychological consultation	✓	✓	
Diagnosis	✓	✓	
Treatment and care according to the mhGAP-IG	✓	✓	
Psychosocial care	✓	✓	✓
Referral/counter-referral	✓	✓	
Preventive services			
Patient follow-up	✓	✓	✓
Reintegration	✓	✓	✓
Promotional services			
Home visits			✓
Mental health awareness and education	✓	✓	✓
Psycho-education	✓	✓	✓
Reporting	✓	✓	✓
Follow-up and supervision	✓	✓	✓
Rehabilitation services			
Support for professional reintegration and reorientation	✓	✓	✓
Occupational therapy	✓		

✓: activity integrated into primary care facility.

**Table 6 ijerph-22-00457-t006:** Profile of health personnel trained in mental health available in Tshamilemba and Mangembo health districts in 2021 and 2022, respectively.

Professional Categories of Trained and Recruited Healthcare Staff	Programme U		Programme R		Total	
	Number	Trained	Number	Trained	Number	Trained
Primary care providers						
Doctor	15	8	10	5	25	13
Nurse	17	12	147	17	164	29
Midwife	2	2	n/d	n/d	2	2
Non-health community stakeholders						
Community health workers	4	4	238	23	242	27
Religious/pastoral/spiritual healer	n/d	n/d	n/d	2	n/d	2
Mental health professionals						
Clinical psychologist	2	2	n/d	n/d	2	2
Social worker	1	1	n/d	n/d	1	1

n/d: no data.

**Table 7 ijerph-22-00457-t007:** Indicators of the use of curative services, including for mental health reasons, in Tshamilemba and Mangembo district health centres from 2022/23 to 2024.

Curative Consultation Utilisation Indicators	2021–2022 ^a^	2022–2023 ^a^	2023–2024 ^a^
Programme U			
1. Estimated Agétraf health area population	12,000	12,360	12,731
2. Number of new cases at curative consultation ^b^	4560	5068	5296
3. Rate of use of general curative consultation (health centre that has integrated mental health services provision) (NC/inhab/year) (=2/1)	0.38	0.41	0.42
4. Number of new cases (NC) with common MNS disorders who visited the general curative consultation at the health centre	85	161	194
5. Proportion of consulted cases for common MNS disorders out of the total use of general curative consultation (=4/2)	1.86%	3.18%	3.66%
6. Rate of general curative consultation for MNS disorders (NC/1000 inhab/year) (=4/1)	7.08‰	13.03‰	14.37‰
Programme R			
1. Estimated population of Mangembo health district ^c^	n/d	40,209	41,415
2. Number of new cases at the curative consultation ^b^	n/d	16,053	17,859
3. Rate of use of general curative consultation (all health centres having integrated the mental health services provision) (NC/inhab/year) (=2/1)	n/d	0.40	0.43
4. Number of new cases (NC) with common MNS disorders who visited the general curative consultation at health centres	n/d	377	586
5. Proportion of cases consulted for MNS disorders out of total use of general curative consultations (=4/2)	n/d	2.35%	3.28%
6. Rate of general curative consultation for MNS disorders (NC/1000 inhab/year) (=4/1)	n/d	9.38‰	14.15‰

^a^ The study year runs from July of year X to June of year Y.; ^b^ Based on our calculation, ^c^ data for half of the health areas that included mental health, the total number of people (N) was divided by 2.

**Table 8 ijerph-22-00457-t008:** Indicators of mental health services use in the Mangembo health district from 2023 to 2024.

Indicators of Mental Health Service Use at the Hospital in the Programme R	2022–2023 ^a^	2023–2024 ^a^
1. Total population of the Mangembo Health District	80,417	82,830
2. Number of admissions to the mental health department of Mangembo Hospital	106	210
3. Rate of hospitalisation for MNS disorders per 1000 people (=2/1)	1.32‰	2.54‰

^a^ The study year runs from July of year X to June of year Y.

**Table 9 ijerph-22-00457-t009:** The proportion (%) of people receiving mental health services in healthcare facilities in the districts of Tshamilemba and Mangembo from July 2021/22 to June 2024.

Programmes	Integrated MNS Disorders	Proportion of People Cared for at Health Centres				Proportion of People Cared for in the District Hospital		
		2021–2022	2022–2023	2023–2024	Total	2022–2023	2023–2024	Total
Programme U		*n* = 85	*n* = 161	*n* = 183	*n* = 429			
	Depression	8.2%	32.3%	38.8%	26.4%	n/a	n/a	n/a
	Psychoses	14.1%	18.0%	16.9%	16.3%	n/a	n/a	n/a
	Epilepsy	3.6%	21.1%	28.4%	17.7%	n/a	n/a	n/a
	CMH	0%	1.2%	4.9%	2.0%	n/a	n/a	n/a
	Dementia	1.2%	0%	1.1%	0.8%	n/a	n/a	n/a
	Substance use disorders	5.9%	8.1%	7.7%	7.2%	n/a	n/a	n/a
	Self-harm/suicidal behaviour	2.4%	1.9%	0.5%	1.6%	n/a	n/a	n/a
	Other significant mental health complaints	64.7%	17.4%	1.7%	27.9%	n/a	n/a	n/a
Programme R			*n* = 377	*n* = 586	*n* = 963	*n* = 106	*n* = 210	*n* = 316
	Depression	n/d	50.1%	35.4%	42.8%	39.6%	30.5%	35.0%
	Psychoses	n/d	24.9%	16.4%	20.7%	28.3%	21.4%	24.9%
	Epilepsy	n/d	22.3%	24.2%	23.3%	30.2%	19.5%	24.9%
	CMH	n/d	0.5%	3.6%	2.0%	0%	4.3%	2.1%
	Dementia	n/d	0%	0.2%	0.1%	0%	0%	0%
	Substance use disorders	n/d	2.1%	9.6%	5.8%	1.9%	9.0%	5.5%
	Self-harm/suicidal behaviour	n/d	0%	0%	0%	0%	0%	0%
	Other significant mental health complaints	n/d	0%	10.6%	5.3%	0%	15.2%	7.6%

CMH: Mental and behavioural disorders in children and adolescents; n/a: not applicable because in Programme U, mental health services were not integrated into the district hospital; n/d: no data available, because Programme R had not yet started.

**Table 10 ijerph-22-00457-t010:** SWOT analysis of integration programmes testing the mhGAP in the DRC context.

Strengths	Weaknesses
Action-research as an integration approach adopted in programme U; presence of a motivated multidisciplinary team; support from development partners and specialists; availability of adequate funding in programme R; awareness-raising in the community; involvement of the entire healthcare team; good collaboration within the team; integration at both levels of care with a functional referral/counter-referral system; presence of sensitised frontline care providers; strong enthusiasm and involvement on the part of health system stakeholders.	Unavailability of psychotropic drugs or insufficient stocks of psychotropic drugs in healthcare facilities; low rate of drug cost recovery, drug stock-outs, and lack of a budget line dedicated to renewing these stocks; low motivation of community health workers; increased workload (tasks and responsibilities) for healthcare providers, supervisors, and, above all, community health workers; mental healthcare expenditure fully covered by households, with 100% out-of-pocket payment.
**Opportunities**	**Threats**
Availability of overall financial support in one health district; assignment of a frontline clinical psychologist; development of partnerships with higher education and university establishments; commitment by the State, through its PNSM, to take ownership of efforts to perpetuate the integrated mental healthcare model in the PHC system developed in the experimental health districts; and support from the Catholic Church, through its not-for-profit associations such as Les Amis de Mangembo, for integration initiatives.	Disengagement of technical and financial partners; weight of beliefs, customs and socio-cultural practises surrounding mental disorders; absence of a social protection system for people with mental disorders; lack or weakness of a policy to retain health personnel trained in mental health; lack or inadequacy of funding for the implementation of integration; scaling up challenges in the absence of sufficient funding.

## Data Availability

All relevant data relating to this study are included in this manuscript. Additional information regarding this study may be obtained upon reasonable request to the corresponding author.

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
