# Peer review of "Integrating Mental Health Services into Primary Care Settings: A Multiple Case Study of Congolese Experiences Testing the Feasibility of the WHO’s Mental Health Gap Action Programme"

_ijerph, 2025, doi:10.3390/ijerph22030457_

Round 1
Reviewer 1 Report
Comments and Suggestions for Authors
This is an important paper, describing two pilots for mhGAP-informed service reform in DRC. The detailed by succinct introduction nicely summarised the huge gaps in mental health care in DRC, and how some of the recent developments in the field of global mental health have been applied.
There is value in formally evaluating such programmes, to inform replication, and making a case for further investment to start to close the treatment gap in DRC, and to understand how mhGAP and associated service reforms work in practice in this particular context.
Table 1 provides a clear and useful summary of the characteristics of the two sites, while it is useful to have the country-level data, it might also have been helpful to also try to compare to an average resource availability in districts (this might be more meaningful than, for example comparing the number of PHC facilities in the whole country).
The multiple-case studies approach is appropriate for the aims of the study, though given the fact that the services are still running (line 757/758) more robust prospective designs with specific outcomes of interest being measured (rather than relying on secondary quant data) would have been ideal. This might be possible (in a sample of patients) in the event of expansion of services, or new pilots. I noted that establishing an integrated health information system was one of the elements of the programmes that was not successful. This is a common challenge, and one that can make good evaluation and learning more difficult. Despite this, the intervention appears to have been very comprehensive, and the amount of data collected was high, making the results meaningful. It is unusual to see so many mhGAP priority conditions included in such programmes. Similarly provision of 11 drugs types is unusually high.
The methods and analysis are clearly described and the triangulation of qualitative and quantitative data is more likely to lead to robust results and conclusions. That said, with secondary quantitative data, there is a tendency towards shorter-term output data (number of people trained), and it is hard to make assertions as to the ongoing impact of these investments. The qualitative interviews give some useful insights into this key issues, and some examples, like medication availability also demonstrate what can happen when funding ceases.
The assertion (line 627) that 75% of people recovered is probably not supported by the data available, as longer-term follow-up information is not available.
Use of the term 'curative consultation' might be unhelpful in implying that the result was of a cure, rather than simply meaning that treatment was provided.
The description of increase in consultations is a sensible approach, though coverage is the preferred metric. I recognise that this can be unhelpful where service provision is so low, but some discussion of how to most usefully measure impact would be interesting in the discussion section. The challenge of not being able to know where people came from (and therefore being able to calculate coverage) is a common one.
In Table 7, it is not clear what is meant by 1. Health area population. This cannot be the total population, comparing to the number of new cases seen. This is carried through into describing eg 38NC/100 inhabitants. Surely over 1/3 of all inhabitants did not seek consultation in a given year as new cases.
It is interesting to read that 100% of people received psychological interventions (line 450), please describe the nature of these interventions, as this is high.
The lessons learned are very appropriate and would be a valuable resource for others considering systems reform in future.
The discussion is a helpful appraisal of the findings, including, for example an open discussion about the quality of the data, and how collection in routine care can lead to major gaps.
The discussion about high rates of depression identified is interesting, in light of the usual profile of service users in similar programmes often being strongly biased towards psychosis and epilepsy, and people with depression being challenging to identify and bring to treatment services.
Overall, this paper uses a good range of data, which clearly outlines the successes and failures of important pilots, which could be a valuable resource for future action.
Author Response
Comment 1: Table 1 provides a clear and useful summary of the characteristics of the two sites, while it is useful to have the country-level data, it might also have been helpful to also try to compare to an average resource availability in districts (this might be more meaningful than, for example comparing the number of PHC facilities in the whole country).
Response 1: In fact, we had no real intention of making such comparisons because these are two health districts with very different profiles. In Table 1, we simply wanted to describe the main characteristics of the districts. As the country data seems to be confusing, we have removed this column.
Comment 2: The multiple-case studies approach is appropriate for the aims of the study, though given the fact that the services are still running (line 757/758) more robust prospective designs with specific outcomes of interest being measured (rather than relying on secondary quant data) would have been ideal. This might be possible (in a sample of patients) in the event of expansion of services, or new pilots. I noted that establishing an integrated health information system was one of the elements of the programmes that was not successful. This is a common challenge, and one that can make good evaluation and learning more difficult. Despite this, the intervention appears to have been very comprehensive, and the amount of data collected was high, making the results meaningful. It is unusual to see so many mhGAP priority conditions included in such programmes. Similarly provision of 11 drugs types is unusually high.
Response 2: As we understand it, our reviewer’s comment raises two concerns: i) prospective design should be favoured over multiple case study design; and ii) both programmes were ambitious in nature, as they sought to integrate all seven of the common MNS disorders of the mhGAP and to provide 11 types of drugs. We have addressed these concerns separately in the manuscript, namely in the Study Limitations section, and in the Discussion section.
i) Regarding the first concern, we mentioned the following:
“However, as multiple case study designs do not allow causal links to be measured, further research would ideally adopt more robust prospective designs with specific outcomes of interest being measured across different stakeholders.” Lines 822-825
ii) For the second concern, we mentioned the following: “The ambitious nature of the two programmes, which aimed to treat the seven common MNS disorders and provide up to 11 types of medication, but which were not commensurate with the financial and logistical resources deployed, would have led to minimal outcomes. We therefore suggest being very pragmatic in the choice of priorities when an integration programme is being considered and, for the two programmes currently underway, relegating certain MNS disorders that are infrequent in the context to the background, to devote the necessary energy to the disorders that are really a priority.” Lines 812-818
Comment 3: The methods and analysis are clearly described and the triangulation of qualitative and quantitative data is more likely to lead to robust results and conclusions. That said, with secondary quantitative data, there is a tendency towards shorter-term output data (number of people trained), and it is hard to make assertions as to the ongoing impact of these investments. The qualitative interviews give some useful insights into these key issues, and some examples, like medication availability also demonstrate what can happen when funding ceases.
Response 3: We agree. We mentioned it in the ‘Conclusions’ sub-section of the study, asking that it be considered in the capitalisation activities planned in the two programmes.
The following sentence was added: “For the moment, everything seems working well to the satisfaction of the mhGAP operations team. However, we remain cautious about the sustainability of the longer-term impacts of these programmes once external funding has ended. It would be useful to examine whether these achievements will be maintained in the longer term during the planned capitalisation activities and final evaluation.” Lines 869-873
Comment 4: The assertion (line 627) that 75% of people recovered is probably not supported by the data available, as longer-term follow-up information is not available.
Response 4: We agree. We preferred to refer to “rates of partial or total remission of symptoms” instead of “recovery rates” reported by healthcare providers. We also indicated the following in the Discussion section: “Primary care providers report high partial or total remission rates (74%), but without providing clear objective criteria for this short-term remission. Conducting surveys involving patients undergoing treatment to obtain long-term follow-up data for these patients would make it possible to confirm (or not) these reported remission and/or recovery rates.” Lines 710-714
Comment 5: Use of the term 'curative consultation' might be unhelpful in implying that the result was of a cure, rather than simply meaning that treatment was provided.
Response 5: Thank you for this comment. Indeed, the indicator ‘curative consultation rate’ refers to patients who used the health centre for an episode of illness, in this case an episode of mental, neurological and substance use disorder; therefore, it does not refer to the reported recovery rate, which is another indicator.
Comment 6: The description of increase in consultations is a sensible approach, though coverage is the preferred metric. I recognise that this can be unhelpful where service provision is so low, but some discussion of how to most usefully measure impact would be interesting in the discussion section. The challenge of not being able to know where people came from (and therefore being able to calculate coverage) is a common one.
Response 6: We agree this comment. Indeed, this limitation has already been mentioned in the Study limitations sub-section. The sentence states: “First, with the available data, it was not possible to analyse the situation by distinct geographical health areas and establish where the patients using the Tshamilemba health centre actually come from.”
Comment 7: In Table 7, it is not clear what is meant by 1. Health area population. This cannot be the total population, comparing to the number of new cases seen. This is carried through into describing eg 38NC/100 inhabitants. Surely over 1/3 of all inhabitants did not seek consultation in a given year as new cases.
Response 7: Thank you for the comment. In the DRC, a health centre is responsible for the population in the health area in which it is located. This is known as the population of responsibility. The general curative consultation rate indicator (38NC/100 inhabitants) refers to all health problems, including mental health problems in particular. We are also counting an episode of illness here. This means that a person who has been admitted for a health problem X at a given time (T1), and who is admitted again for a health problem Y at another given time (T2) is considered to have two different cases in the calculation of this indicator.
Comment 8: It is interesting to read that 100% of people received psychological interventions (line 450), please describe the nature of these interventions, as this is high.
Response 8: These were basic psychological interventions. With the recruitment–at the start of programme U, of two clinical psychologists at the health centre of Tshamilemba, all patients systematically received basic psychological interventions. We have clarified the corresponding statement as follows: “Of the people treated under programme U, 100% received psychological interventions (such as psycho-education and counselling to all patients, or interpersonal therapy, cognitive-behavioural therapy, etc. to others, depending on their mental condition), …” Lines 463-464
Comment 9: The lessons learned are very appropriate and would be a valuable resource for others considering systems reform in future.
Response 9: Thank you very much.
Comment 10: The discussion is a helpful appraisal of the findings, including, for example an open discussion about the quality of the data, and how collection in routine care can lead to major gaps.
Response 10: Thank you very much
Comment 11: The discussion about high rates of depression identified is interesting, in light of the usual profile of service users in similar programmes often being strongly biased towards psychosis and epilepsy, and people with depression being challenging to identify and bring to treatment services.
Response 11: Thank you very much.
Comment 12: Overall, this paper uses a good range of data, which clearly outlines the successes and failures of important pilots, which could be a valuable resource for future action.
Response 12: Thank you very much.
Reviewer 2 Report
Comments and Suggestions for Authors
This is an interesting paper on a very important topic of integrating mental health services into primary health care settings in Congo.
Below are comments that are aimed at improving the paper.
Abstract
Generally, it is not advisable to use acronyms/abbreviations in the abstract, particularly U and R.
In lines 19 – 22, the authors mentioned that “A multiple case study was conducted on two integration 19 experiences in the urban Tshamilemba district (2021-25) in the Haut-Katanga province and in the 20 rural Mangembo district (2022-26) in the Kongo Central province, called Programmes U (for Urban) 21 and R (for Rural), respectively.” However, in the following sentence, they mentioned that “Data were collected between July and August 2024..”. The years mentioned in these sentences are confusing unless the authors correct the first sentence to reflect that these programmes are ongoing.
Introduction
I would suggest the authors provide the year(s) mhGAP became policy and approach as mentioned in lines 91 – 92. to add more literature review on recent HIV epidemic statistics.
From line 105, DGD is written differently compared to the Table 2 description. For example, “Belgian” and “ and Humanitarian Aid” are missing in the introduction.
Authors must give more details regarding the national sub-sectoral mental health policy mentioned in lines 120 – 121, particularly the year the policy was enacted.
Materials and Methods
Lines 145 – 150 must be properly referenced.
“One district hospital in Mangembo” is missing in Table 1.
I found Table 1 confusing (e.g. characteristics 4 & 5). Authors should consider adding rows or spaces between variables.
Section 2.4 and Tables 3, 4 & 5 should be presented as results.
In general, I found the interchangeable use of Tshamilemba, CSART, and Programme U within the manuscript to be quite confusing. Similarly, regarding the use of Mangembo, BDOM, and Programme R.
Whilst we appreciate the details of the experts from Guinea and Belgium, the authors must also provide the details of the expert from DRC.
Results
I would suggest that authors should add a row with totals in Table 6, both for number and trained for both programmes.
Also, in Table 6, some numbers and trained are missing. Authors should consistently indicate when there is no data/n/d.
From line 400, nil is explained differently compared to the remarks in Table 3. I found the remark that “there is no data available” more appropriate. However, the authors should cross-check and explain the missing numbers from Table 6, 7 (under Programme R) and 9 under Programme U - Proportion of people cared for in Mangembo hospital and under Programme R - Proportion of people cared for at health centres (2021 - 2022).
The authors mentioned that they conducted 4 individual interviews with public (mental) health experts, but in the results, they only mentioned 1 in line 426. Nevertheless, the description from line 418 suggests that the quotation was from the FDGs. Also, authors should acknowledge this limitation.
Author Response
Abstract
Comment 1: Generally, it is not advisable to use acronyms / abbreviations in the abstract, particularly U and R.
Response 1: We agree. We have deleted the acronyms and replaced them with “the Tshamilemba district” and “the Mangembo district”. Lines 28-29
Comment 2: In lines 19 – 22, the authors mentioned that “A multiple case study was conducted on two integration 19 experiences in the urban Tshamilemba district (2021-25) in the Haut-Katanga province and in the 20 rural Mangembo district (2022-26) in the Kongo Central province, called Programmes U (for Urban) 21 and R (for Rural), respectively.” However, in the following sentence, they mentioned that “Data were collected between July and August 2024.”. The years mentioned in these sentences are confusing unless the authors correct the first sentence to reflect that these programmes are ongoing.
Response 2: We agree. We have specified that the Tshamilemba programme has been underway since 2021, and the Mangembo programme has been in place since 2022. We rephrased the sentence as follows: “A multiple case study was conducted on two integration programmes in the urban Tshamilemba district in the Haut-Katanga province, ongoing since 2021, and in the rural Mangembo district in the Kongo Central province, ongoing since 2022.” Lines 19-21
Introduction
Comment 3: I would suggest the authors provide the year(s) mhGAP became policy and approach as mentioned in lines 91 – 92. to add more literature review on recent HIV epidemic statistics.
Response 3: Thank you for this amendment. For the first part of the amendment, we have corrected this statement: “The mhGAP is a guideline developed in 2008” and not “a policy or approach”. Lines 92
For the second part of the amendment, (... complete the literature review on recent statistics on the HIV epidemic), we feel that we have misunderstood. Indeed, in this manuscript, mental health is deliberately taken in a broad sense, without any particular focus on any one illness.
Comment 4: From line 105, DGD is written differently compared to the Table 2 description. For example, “Belgian” and “and Humanitarian Aid” are missing in the introduction.
Response 4: We agree. The sentence on line 105 in the introduction was an omission. We have included the missing words “Belgian” and “Humanitarian Aid”. Lines 106-107
Comment 5: Authors must give more details regarding the national sub-sectoral mental health policy mentioned in lines 120 – 121, particularly the year the policy was enacted.
Response 5: We provided some essential details on the national sub-sectoral mental health policy. The following paragraph was added:
“This sub-sectorial mental health policy in the DRC was formulated in 1999 and revised in 2021 [12]. It aims to promote the mental health of the entire Congolese population by providing comprehensive, integrated and continuous quality health care and services with community participation, in the overall context of the fight against poverty. Its vision is to ensure that all Congolese have equitable and sustainable access to quality mental health services and care defined in the (minimum and complementary) packages of activities, adjusted to the needs of users, taking account of contextual changes. The sub-sectoral metal health policy comprises seven strategic orientations: i) the integration of mental health services and care packages into the health district and the continuity of care; ii) the strengthening of leadership and governance; iii) the development of human resources involved in mental health; iv) the improvement of infrastructures and the availability of mental health equipment and materials; v) the improvement of the availability of psychotropic medicines and supplies; vi) the improvement of mental health funding; and vii) the strengthening of intra- and inter-sectoral partnerships [12].” Lines 124-137
Materials and Methods
Comment 6: Lines 145 – 150 must be properly referenced.
Response 6: OK. We have added references.
Comment 7: “One district hospital in Mangembo” is missing in Table 1.
Response 7: Thank you for this amendment. Mangembo hospital is one of the 18 primary care facilities mentioned in Table 1, in the column relating to Mangembo district. In the DRC, health centres and district hospitals are all part of primary care facilities.
Comment 8: I found Table 1 confusing (e.g. characteristics 4 & 5). Authors should consider adding rows or spaces between variables.
Response 8: Thank you for the comment. We have modified table 1 by including characteristic 5 in 4 and then 7 in 6, expressing them as percentages. We also added spaces to distinguish between the characteristics.
Comment 9: Section 2.4 and Tables 3, 4 & 5 should be presented as results.
Response 9: We have moved tables 3, 4 and 5 and the related comments to the Results section.
Comment 10: In general, I found the interchangeable use of Tshamilemba, CSART, and Programme U within the manuscript to be quite confusing. Similarly, regarding the use of Mangembo, BDOM, and Programme R.
Response 10: Thank you for the comment. We have removed the two acronyms from the manuscript, namely: ”CSART” to refer directly to the health centre of Tshamilemba, and ”BDOM” to refer directly to the Mangembo district.
Comment 11: Whilst we appreciate the details of the experts from Guinea and Belgium, the authors must also provide the details of the expert from DRC.
Response 11: Thank you for this amendment. Normally, the details of the DRC experts are typically presented in the Results section. To take this amendment into account and avoid repetition, we have moved the details of the Belgian and Guinean experts to the Participant Characteristics sub-section in the Results section.
Results
Comment 12: I would suggest that authors should add a row with totals in Table 6, both for number and trained for both programmes.
Response 12: Thank you very much. We have added totals to table 6.
Comment 13: Also, in Table 6, some numbers and trained are missing. Authors should consistently indicate when there is no data/n/d.
Response 13: Thank you very much. We mentioned that there is no data available (n/d).
Comment 14: From line 400, we can see that nil is explained differently compared to the remarks in Table 3. I found the remark that “there is no data available” more appropriate. However, the authors should cross-check and explain the missing numbers from Table 6, 7 (under Programme R) and 9 under Programme U - Proportion of people cared for in Mangembo hospital and under Programme R - Proportion of people cared for at health centres (2021 - 2022).
Response 14: We have replaced: ”nil” with ”unknown”. Lines 411, 416
For tables 6, 7, and 9, we have indicated: ”no data” where there is no data available, and ”not applicable” where the programme had not yet begun (2021-2022), especially for the programme R. See Tables 6, 7 and 9
Comment 15: The authors mentioned that they conducted 4 individual interviews with public (mental) health experts, but in the results, they only mentioned 1 in line 426. Nevertheless, the description from line 418 suggests that the quotation was from the FDGs. Also, authors should acknowledge this limitation.
Response 15: The source of the result on line 426 was incorrect. We have corrected this. The public (mental) health experts were especially more involved in: i) SWOT (strengths, weaknesses, opportunities and threats) analysis of the programmes; and ii) drawing the lessons learned. The way in which we have chosen to present these results does not favour direct indication of the source of the response. That is why we don't see the results of individual interviews.
Reviewer 3 Report
Comments and Suggestions for Authors
Thanks for the opportunity to review this manuscript.
Integrating mental health into primary health services is a topic that I’m personally interested in and it is taking attention globally at the moment.
- In line 53, you mentioned that the mental health coverage is only 5 %, and then in line 64, it was mentioned that only 2 of 10 people have access to mental health services. What is the difference between both numbers? Which one is right?
- Line 72, the paragraph starts with “However PHC setting..” needs to be connected to the other sentences as this can not be the beginning of a paragraph. The same for the paragraph starting at line 87 (it can not start with then).
- Line 91: I would argue that mhGAP is not a policy. It is guidance of how mental health could be integrated into PHC.
- Line 92: mhGAP intervention guide does not describe the integration phases, the operations manual does.
- Line 116 – Box 1: the three phases described in the mhGAP operations manual are: plan, prepare, and provide. Please review the operations manual and edit accordingly. Also, it is better to present this as bullet points, not as one paragraph.
- Please change mhGAP-GI to mhGAP-IG as it stands for intervention guide.
- Line 189: please define clearly who were trained on mhGAP. What is meant by non-specialist and non-professional ?
- Line 255: it should be quantitative outcomes, not statistical outcomes, as no statistical analysis was conducted. Similarly, line 266, it should be quantitative data
- Define clearly the research outcomes you are assessing and how the feasibility of mhGAP would be measured, i.e. what are the indicators of feasibility
The quality needs to be improved. I have included some suggestions in the feedback.
Author Response
Comment 1: In line 53, you mentioned that the mental health coverage is only 5 %, and then in line 64, it was mentioned that only 2 of 10 people have access to mental health services. What is the difference between both numbers? Which one is right?
Response 1: As the data source for the second estimate ”only 2 in 10 people have access to mental health services” did not provide us with any further details, we preferred to use the first figure (Line 53), which comes from an official source. The sentence on line 64 has therefore been reworded as follows: ”the majority of people in the DRC do not have access to evidence-based mental health services.” Lines 64-65
Comment 2: Line 72, the paragraph starts with “However PHC setting.” needs to be connected to the other sentences as this cannot be the beginning of a paragraph. The same for the paragraph starting at line 87 (it cannot start with then).
Response 2: The sentence on line 72 has been reworded as follows: ”In the DRC, PHC settings still have some way to go before they can offer quality services, including mental health services.” Lines 72-73
The sentence in line 82 has also been reworded as follows: ”To facilitate the implementation of integration in resource-limited countries, WHO [19] has developed the Mental Health Gap Action Programme (acronym ‘mhGAP’).” Lines 87-88
Comment 3: Line 91: I would argue that mhGAP is not a policy. It is guidance of how mental health could be integrated into PHC.
Response 3: We agree. We have corrected this throughout the text. The words ”policy” and ”approach” applied to mhGAP have either been replaced by ”guideline or guidance” or deleted where they were not required to be specified.
Comment 4: Line 92: mhGAP intervention guide does not describe the integration phases, the operations manual does.
Response 4: Thank you very much. We have corrected the sentence as follows: ”The mhGAP operations manual describes the three-phase mental health integration process and its main activities, which are listed below (Box 1).” Lines 93-94
Comment 5: Line 116 – Box 1: the three phases described in the mhGAP operations manual are: plan, prepare, and provide. Please review the operations manual and edit accordingly. Also, it is better to present this as bullet points, not as one paragraph.
Response 5: Thank you very much. We have corrected the phases and activities in Box 1. Lines 118-119
- Plan, which includes activities such as: setting up an mhGAP operational team, conducting a situation analysis, developing an mhGAP operational plan and budget, and advocating for mental health;
- Prepare, which includes activities such as: adapting the components of the mhGAP package, training staff in mhGAP, preparing for clinical and administrative supervision, coordinating care pathways, improving access to psychotropic drugs and improving access to psychological interventions;
- Provide, which includes activities such as: providing services at facility level, providing treatment and care in the community, raising awareness of MNS conditions and available services, and supporting the implementation of prevention and promotion programmes.
Comment 6: Please change mhGAP-GI to mhGAP-IG as it stands for intervention guide.
Response 6: Thank you very much. We have replaced ”mhGAP-GI” with “mhGAP-IG” throughout the manuscript.
Comment 7: Line 189: please define clearly who were trained on mhGAP. What is meant by non-specialist and non-professional?
Response 7: These were primary care providers (who are not mental health specialists) and community players (non-health professionals) whose profiles are shown in Table 4. We have made this clarification:
”In both programmes, primary care providers (non-specialists in mental health) and other community stakeholders (non-health professionals) were trained in the modules defined in the mhGAP-IG and in other essential concepts on mental health and the integration of services. Table 4 presents the content of the training received by primary care providers and other community stakeholders.” Lines 316-320
Comment 8: Line 255: it should be quantitative outcomes, not statistical outcomes, as no statistical analysis was conducted. Similarly, line 266, it should be quantitative data
Response 8: Thank you very much. We have corrected this by replacing “statistical outcomes” with “quantitative outcomes” and “statistical data” with “quantitative data”. Lines 222, 234
Comment 9: Define clearly the research outcomes you are assessing and how the feasibility of mhGAP would be measured, i.e. what are the indicators of feasibility
Response 9: Thank you for the comment. As we stated in the Data analysis sub-section, to ensure that the process was feasible in a context where mental health services were not available despite growing demand, we measured a number of health indicators such as: Mental health services access indicators (e.g., availability of human resources and psychotropic drugs, geographical accessibility, financial accessibility, etc.). Functioning of (mental) health services utilisation indicators (Rate of use of curative mental health services in general curative consultations, general curative consultation for common MNS disorders rate, hospitalisation rate for common MNS disorders, etc.). the opinions of the stakeholders expressed during the focus groups and interviews regarding SWOT analysis of the programmes, as well as the lessons learned, allow us to say that it is feasible to integrate mental health services using the mhGAP.
In this study, “testing feasibility” did not refer to the use of statistical tests to compare the scores obtained for indicators measured at baseline (T0) and T1 of the study. This exercise is planned for the next stages of the main research project.
Comment 10: The quality needs to be improved. I have included some suggestions in the feedback
Response 10: Thank you for the comment. We have taken this into account, and the manuscript has been proofread by a native English speaker.
Round 2
Reviewer 3 Report
Comments and Suggestions for Authors
Thanks for responding to my comments.
I have no further comments.